# Occurrence of Reading Skills in a National Age Cohort of Norwegian Children with Down Syndrome: What Characterizes Those Who Develop Early Reading Skills?

**DOI:** 10.3390/brainsci11050527

**Published:** 2021-04-21

**Authors:** Kari-Anne B. Næss, Egil Nygaard, Elizabeth Smith

**Affiliations:** 1Department of Special Needs Education, University of Oslo, 0318 Oslo, Norway; 2Department of Psychology, University of Oslo, 0317 Oslo, Norway; egilny@psykologi.uio.no; 3School of Psychology, University of Cardiff, Cardiff CF10 3AS, UK; SmithL57@cardiff.ac.uk

**Keywords:** trisomy 21, decoding, vocabulary, letters, phonological awareness

## Abstract

Children with Down syndrome are at risk of reading difficulties. Reading skills are crucial for social and academic development, and thus, understanding the nature of reading in this clinical group is important. This longitudinal study investigated the occurrence of reading skills in a Norwegian national age cohort of 43 children with Down syndrome from the beginning of first grade to third grade. Data were collected to determine which characteristics distinguished those who developed early reading skills from those who did not. The children′s decoding skills, phonological awareness, nonverbal mental ability, vocabulary, verbal short-term memory, letter knowledge and rapid automatized naming (RAN) performance were measured annually. The results showed that 18.6% of the children developed early decoding skills by third grade. Prior to onset, children who developed decoding skills had a significantly superior vocabulary and letter knowledge than non-readers after controlling for nonverbal mental abilities. These findings indicate that early specific training that focuses on vocabulary and knowledge of words and letters may be particularly effective in promoting reading onset in children with Down syndrome.

## 1. Introduction

Reading is the cognitive process of decoding words and obtaining meaning from text [1]. Functional reading skills expand opportunities for learning and participation in educational and social activities at home, in school, at work and in society [2,3], substantially impacting individuals’ lives. Due to their reduced cognitive capacity, children with Down syndrome often experience reading difficulties [4]. In recent decades, however, reading skills and other academic achievements have improved among children with Down syndrome [5]. These improvements could be due to the higher educational goals being set for them and their increased inclusion in mainstream education, where literacy is a key area of the curriculum [5,6,7]. However, the proportion who successfully develop reading skills in their early school years is unknown, and which early cognitive skills promote early reading success is unclear.

Understanding the development of functional reading skills in children with Down syndrome in the lead-up to entering primary school is crucial for planning early educational interventions. Therefore, this study aims to investigate the occurrence of reading skills in children with Down syndrome and to provide insight into early abilities prior to reading onset and formal education, as these abilities may differ between young readers and non-readers with Down syndrome.

### 1.1. Development of Reading

For beginning readers, much energy is dedicated to the technical part of the reading process to learn to decode; however, as reading becomes more fluent, children dedicate more energy to linguistic comprehension (cf. automaticity theory; [8,9]). In stage models describing technical development, children begin with a logographical strategy (sight word reading without knowing the alphabetical principle), followed by a phonological decoding strategy (phoneme–grapheme correspondence is gradually established, and phonemes are synthesized into syllables and words) and then an orthographic decoding strategy (the orthographic, phonetic and semantic identities of words are stored in long-term memory and can be directly accessed). Over time, increased print experience results in increased automaticity and fluent decoding (for an overview, see Frith [10]). Typically developing children become fluent readers at approximately 3rd grade [11].

Different stage models exist (see also, e.g., [12]), and they have been criticized for oversimplifying the process of decoding development (e.g., [13]), overlooking, for example, word familiarity, word complexity, whether a word appears alone or in context [14] and the transparency of language [15]. However, such models provide a general framework for understanding how children transition from one decoding strategy to the next (e.g., [16]), and they emphasize that there are different subskills of word identification [17].

### 1.2. Development of Reading in Children with Down Syndrome

It has been suggested that rather than progressing through the above stages in decoding development, children with Down syndrome tend to rely on logographic strategies throughout their school years and beyond [18]. Reliance on such strategies may explain the difficulties with non-word decoding observed in groups with Down syndrome [19,20]. However, some children with Down syndrome develop exceptional reading skills [21], and a small proportion perform in line with typically developing peers of the same chronological age [19]. Thus, in individuals with Down syndrome, poor decoding skills are certainly not inevitable.

### 1.3. The Occurrence of Reading in Children with Down Syndrome

Differences in school placement, access to intervention, how reading is taught and expectations about children’s potential may lead to very different reading outcomes among children with Down syndrome. Thus, as Groen et al. [21] note, it is difficult to know what level of reading ability to expect at a given point in time. Additionally, longitudinal research investigating the occurrence of reading skills in children with Down syndrome is limited. However, a five-year longitudinal study by Bird et al. [22] and a two-year longitudinal study by Byrne et al. [23] both included children below the age of 13 years (at the beginning of the study) and reported that 83.3% and 87.5% of participants, respectively, were able to decode words at a measurable level by the end of the study. In a five-year longitudinal study by Laws and Gunn [24], who included a mix of children, adolescents and adults (10–24 years of age at the beginning of the study), there was an occurrence of 53% at the end of the study. Thus, in previous research, the occurrence of reading skills in individuals with Down syndrome varies widely. Notably, there are no cohort studies that provide age-specific occurrence data on the reading ability of children with Down syndrome. The two studies above focusing specifically on the occurrence of reading in children [22,23] include age ranges of 5 years and 8 years, age ranges within which children could be expected to show vast differences in ability [11]. The combination of the wide age ranges and small sample sizes (*n* = 12 [22]; *n* = 24 [23]; *n* = 30 [24]) in the abovementioned studies means that it is not possible to meaningfully break down occurrence by age. These limitations and the lack of cohort studies in the early years mean that we do not know what to expect with regard to children with Down syndrome at specific ages/stages of development, such as the early school years. In an age cohort, the reading abilities across children may be expected to be within a more confined range due to less age- and experience-related variance.

### 1.4. Variables Related to the Development of Reading Skills

To suggest appropriate intervention routes specifically adapted to the phenotype of Down syndrome, it is also critical to explore the variables related to early reading skills in this population. In typically developing children, there is a consensus that phonological skills play a key role; phonological awareness, verbal memory, rapid automatized naming (RAN) and/or letter knowledge have been repeatedly found to predict reading performance [25,26,27]. Notably, the development of phonological awareness has been found to be tied to basic lexical knowledge. Walley et al. [28] suggest that vocabulary growth leads to segmental lexical representations, which are thought to be important for explicit phonemic segmentation and phonemic awareness. Therefore, vocabulary has also been found to positively affect children’s reading development (cf. [29,30]) and to differentiate between typical readers and poor readers [31,32]. The associations among phonological skills, vocabulary and reading may be logical since the decoding process proceeds through the previously presented stages: from visually driven coding between printed letters and word pronunciations to the more sophisticated use of phonological and lexical information aggregating more effective word recognition processes [10,33].

The role of phonological variables in predicting reading skills in children with Down syndrome has been debated. There is a consensus that children with Down syndrome generally have weak phonological skills (letter knowledge [19]; phonological awareness [34]; memory [35]), which may in itself call into question the impact of these skills on reading development and suggest that other variables may have stronger compensatory influences. However, the association between phonological skills and reading skills among children with Down syndrome varies across studies. While some studies have concluded that phonological variables (e.g., phonological awareness [18,36,37,38,39,40], memory [23,41,42], RAN [21], and letter knowledge [42]) play a key role in decoding outcomes, reading has also been observed in this population in the absence of certain phonological skills, e.g., phonological awareness [43]. Notably, in the existing longitudinal studies on reading development in children with Down syndrome, recruitment was conducted after reading onset. Since the relationship between phonological awareness and reading is suggested to be reciprocal in nature, phonological awareness may have promoted early reading, which in turn augmented the development of phonological awareness [44,45]. Therefore, variations in children’s reading experiences may be associated with variations in both the level of mastery of phonological awareness and the strength of its association with reading.

Various studies indicate that language skills, including vocabulary, also play a role in the reading development of individuals with Down syndrome (e.g., [22,24,40]). Notably, Hulme et al. [19] and Boudreau [46] found that language was a stronger predictor of reading ability in children with Down syndrome than in nonverbal, mental-age-matched, typically developing children. For groups with an impaired phonological pathway and weak decoding skills, such as children with Down syndrome [39], semantic word knowledge has been argued to be more important [47,48]. Similarly, familiarity with the spoken form of a new word may be particularly helpful in supporting reading in these children, potentially providing some compensation for these other difficulties.

Finally, since there appears to be a weak but consistent relationship between nonverbal mental ability and general reading skills in typically developing children [49], nonverbal mental ability is also an important variable to consider, as children with Down syndrome usually have intellectual disabilities. Several studies have found indications of such an association. For example, Laws and Gunn [24] found evidence of a significantly higher nonverbal mental ability score in readers than in non-readers with Down syndrome (e.g., [24]). However, because of the low number of participants usually included in studies of children with Down syndrome, the unique contribution of phonological skills over and above nonverbal mental ability has seldom been reported.

In addition to the underlying cognitive skills mentioned above, other variables, such as the home literacy environment [50], socioeconomic status [51] and schooling [34], may impact the reading skills of those with Down syndrome. The effect of hearing on reading development in children with Down syndrome has also been debated (e.g., [19]). However, the present study focuses on understanding which cognitive variables are underlying strengths in children with Down syndrome who develop early reading skills; the findings may indicate which variables enhance reading ability in this population early in development. Supporting reading-associated variables from an early age could provide greater potential for future reading success; the effect of such support is a well-established finding in typically developing children (e.g., see the article on “Matthew effects” by Stanovich [52]) and may also apply to individuals with Down syndrome.

### 1.5. Summary and Research Questions

The occurrence of reading skills in children with Down syndrome across the early school years from the time when formal teaching starts remains unknown. Thus, few studies have provided guidelines on appropriate expectations with regard to reading outcomes and approaches to teaching reading to children with Down syndrome. In all of the abovementioned studies on occurrence, the majority of subjects varied widely in age and were recruited post-reading onset. Thus, the occurrence of reading skills in young children with Down syndrome internationally may not be as high as suggested in the previous literature. If this is the case, it is important to know, as it may influence, e.g., parents’ views on their children’s development and teachers’ expectations.

Furthermore, knowledge of what differentiates those with early decoding skills from those without these skills is limited because the low number of participants in previous longitudinal studies has limited the number of predictive variables included in the analysis.

Therefore, in the present study, we ask the following research questions: (1) What is the occurrence of reading skills in a national age cohort of Norwegian children with Down syndrome in grades 1, 2 and 3 (ages 6–8), and (2) what distinguishes the cognitive profiles of readers and non-readers prior to reading onset? The present study focuses on an age cohort to provide more specific information about occurrence in relation to age to better inform expectations, and it also includes a measure of parental report alongside a standardized measure to draw comparisons between outcomes for these respective measures.

## 2. Materials and Methods

The results reported in this paper are original reading data obtained from a larger research project studying a national age cohort of 43 children with Down syndrome. The title of the project is “Language and reading development in children with Down syndrome” [53].

### 2.1. Participants

A Norwegian national age cohort of six-year-old children with Down syndrome (including every registered child across the country) was invited to participate; all habilitation services in Norway forwarded an informational letter and a consent form to the registered parents of each child with Down syndrome. The letter and consent form were approved in advance by the Regional Committees for Medical and Health Research Ethics. The families of forty-three children with Down syndrome accepted the invitation on the children’s behalf (22 boys and 21 girls; chronological age: mean (*M*) = 75.78 months, *SD* = 3.48 months; nonverbal mental ability raw score (Block Design): *M* = 12.23, *SD* = 5.40). The families who accepted the invitation returned the consent form to the principal investigator. In addition to being 6 years old at the start of the study, the inclusion criteria were that the child did not have a comorbid diagnosis of autism spectrum disorder (ASD) and that Norwegian was the first language.

Among the readers, all children had trisomy 21 except one who had translocation. The non-readers showed almost identical percentages. All except two children had trisomy 21. One of the two had translocation, and the other had mosaic. All participants in both groups except for one of the non-readers went to ordinary primary school. At T1, there were quite similar occurrences between the two groups in regard to permanent hearing disability (25% vs. 30% for non-readers and readers, respectively, odds ratio (*OR*) = 0.77, 95% confidence interval (*CI*) 0.13–4.48, *p* = 0.77) and mean parental education (*M* = 2.51, *SD* = 1.07 vs. *M* = 2.56, *SD* = 1.02 for non-readers and readers, respectively, *OR* = 1.05, 95% *CI* 0.50–2.20, *p* = 0.91). However, at T1, the nonverbal mental ability of readers (*M* = 3.13, *SD* = 2.10) was substantially better than that of non-readers (*M* = 1.49, *SD* = 1.07, *OR* = 1.95, 95% *CI* 1.13–3.41, *p* = 0.01) (scaled scores based on the Block Design; for a description of the measure, see Section 2.3.3).

### 2.2. Data Collection

Data were collected through clinical assessment of the children and through parental questionnaires. The children were assessed every autumn for their first three school years. They were assessed individually in separate rooms in three sessions, typically on consecutive days. All answers were registered manually in the standardized test protocol, and expressive answers were audio recorded for subsequent verification. The parental questionnaire was sent to one parent of each participating child. Up to two reminders were sent if no answers were received by the deadline. The answers were automatically coded in SPSS.

### 2.3. Measures

Standardized procedures for the implementation and scoring of the tests were followed. The tests that were used were originally developed for typically developing children. Although they have not been specifically validated for children with Down syndrome, they have been commonly used in research involving this group of children. Internal consistency, which is a function of the number of test items and the average inter-correlation among the items for the current sample, exhibited reasonably good reliability for all tests (ranging between α = 0.77 and α = 0.95), except for the initial syllable measure (α = 0.57). Notably, for RAN, reliability was calculated by the intraclass correlation between RAN1 and RAN2 and was found to be moderate, *ICC* = 0.57, when using a two-way mixed-effects model with absolute agreement based on an average of the two measures.

#### 2.3.1. Reading Measures

The dependent variable was children’s reading skills, which were assessed using a standardized test for decoding and spelling, STAS-OA-1 [54]. In STAS-OA-1, children are shown a list of high-frequency, phonetically regular single words (without any visual context/support), and they are instructed to read the words aloud for 40 s. The word list starts with two-syllable words, and the length and difficulty of the words gradually increase. Children earn one point for every word read correctly. Spelling is not considered reading; for example, if a child says the letters “c”, “a” and “t” separately, they do not score a point. Children have to synthesize the phonemes into a word to score a point. In this standardized measure, reading reflects the decoding of different word classes without any contextual support, which is essential for effective independent reading. The STAS is used in Norway as a standard reading assessment strategy for all children in mainstream schools, and it has been shown to be highly reliable (e.g., [55]). Children who scored more than 1 point on the test at any time point (first grade: T1, second grade: T2 or third grade: T3) were classified as “readers”. This definition is based on earlier studies (e.g., [24]).

At T1, reading was also measured as part of a large digital parental questionnaire on different background measures categorizing the number of words the child could recognize based on the following scale: 0 words = 0; 1–5 words = 1; 6–10 words = 2; 11–15 words = 3; more than 15 words = 4. There was also a category for unknown.

#### 2.3.2. Background Measures

In the background questionnaire, we also collected information about the types of Down syndrome, school types, permanent hearing loss and the parents’ highest educational level.

For the types of Down syndrome, the response options were 1 = I do not know, 2 = trisomy 21, 3 = translocation and 4 = mosaic. For school types, the response options were 1 = ordinary school, 2 = special school and 3 = other. For permanent hearing loss, the response options were 0 = no and 1 = yes. For the parents’ highest educational level, the response options were elementary school = 0, high school (1–2 years) = 1, high school (3–4 years) = 2, university level up to 3 years = 3 and university level 4 years or more = 4. A mean parental score was calculated based on the average of the mother’s and father′s educational level.

#### 2.3.3. Nonverbal Mental Ability

Nonverbal mental ability was assessed via the Block Design subtest of the third edition of the Wechsler Preschool and Primary Scale of Intelligence (WPPSI-III) [56]. In this subtest, a child is shown several building blocks put together in a pattern either with blocks (items 1–13) or via a picture (items 13–20). The child then has to copy the block arrangement. There are twenty items in total. Two points are scored each time the child correctly copies the block arrangement. For the first six items, the child is allowed two attempts, earning one point if they are correct on the second attempt rather than the first. Specified starting points and discontinuation rules were followed.

#### 2.3.4. Vocabulary

The Norwegian versions of the British Picture Vocabulary Scale (BPVS-II; [57,58]) and Picture Naming (WPPSI-III; [56]) were used to assess vocabulary. For each item in the BPVS-II, a child is shown four pictures and is then asked to point to the picture corresponding to the word spoken aloud by the examiner. The test consists of 144 items, with specified starting points and discontinuation rules. The child earns one point for each correct answer.

The Picture Naming (WPPSI-III; [56]) task involves showing a child a set of single pictures, one item at a time, and then asking them to name the pictures. One point is scored for each correct answer, and no penalties are assessed for articulation errors. The test consists of 38 items, with specified starting points and discontinuation rules.

Vocabulary was calculated as the mean of the z-values of the raw scores on each of the two tests.

#### 2.3.5. Verbal Short-Term Memory

Verbal short-term memory skills were measured via word span, non-word repetition and sentence memory.

In the word span task [59], a child hears a list of spoken words, and their task is to repeat the words in the correct order. The length of the word list gradually increases. The child earns one point for every list repeated correctly and is not penalized for systematic articulation errors. The test consists of 24 items. All the children started on word list 1 and continued until the discontinuation point was reached.

Non-word repetition was assessed using a Norwegian version of the Children’s Test of Non-Word Repetition [60,61]. In each trial, a child hears a non-word, which they have to repeat. The non-words vary in length from two to five syllables. The child earns one point for every correct item and is not penalized for systematic articulation errors. The test consists of 28 items.

Sentence memory was assessed by the Sentence Repetition Test (WPPSI-R; [62]), in which a child listens to spoken sentences that they need to repeat. The child earns one point for every correct item and is not penalized for systematic articulation errors. The test consists of 21 items. All the children continued until a discontinuation point was reached.

Z-values based on the raw scores on the three tests were combined into a measure of mean verbal short-term memory.

#### 2.3.6. Letter Knowledge

The letter sound test from the Aston Index [63] was used to assess letter knowledge. Twenty-four letters are included (c, w, x, z, and q are excluded). The letters are presented in six rows of four letters each. Each time the examiner points to a letter, the child’s task is to decode that letter. The child earns one point for each correct answer, with both letter names and letter sounds accepted as answers. The results presented are z-values based on raw scores.

#### 2.3.7. Rapid Automatized Naming

Two tasks, object RAN tasks for young Norwegian children [59], were used to assess RAN. All words included in the tasks were high-frequency words usually acquired at a very early age. In the first RAN task (RAN1), a child is given a sheet of paper showing black and white drawings of a sun, boat, mouse, door and bus. The five pictures are shown randomly in four rows with five items in each row. The child is asked to name each picture. The number that the child named correctly and the time that it took them to complete the task are recorded. In the second RAN task (RAN2), the child carries out the same task as in RAN1, but the pictures are of a light, ball, boy, house and car. The scoring scheme for RAN2 is the same as that for RAN1. The mean summary scores were calculated for the z-value of the total amount of time used on each of the two tasks.

#### 2.3.8. Phonological Awareness

Four implicit measures of phonological awareness (initial syllable matching, final syllable matching, rhyme matching and initial phoneme matching) and their standardized procedures were adapted from Carroll et al. [64]. In each of the tasks, a child is shown a puppet, such as “Frode the frog”; for the initial syllable matching test, the child is told that the puppet likes to collect words that start with the same syllable. For each item, the frog puppet holds a picture card in front of the child, while two more picture cards are laid on the table. The child is asked to point to the picture beginning with the same syllable. The task was presented in the same way for the other three phonological awareness measures (with a different puppet for each), where children were told that the puppet would like to collect words with the same final syllable, words that rhymed/sounded the same, or words with the same initial sound. For every correct answer across these four measures, the child earns one point, and the summary scores for each of the measures are calculated separately. Both of the syllable measures consist of 8 items, while the remaining two measures consist of 16 items each. The four test results were standardized (z-values) and combined into a mean phonological awareness score.

For all the measures, standardized procedures were followed. Practice examples were provided before the tests started to ensure that the children understood each task.

### 2.4. Analysis

The dependent variable, the STAS-OA1 test, was dichotomized due to skewed results. For the other measures, we have specified in the description above how each of the measures were combined into summary scores. Binary logistic regression analyses, both bivariate and controlled for nonverbal mental ability, were used to analyze the differences between groups. The assumptions for logistic regression analyses were satisfied based on Box–Tidwell tests and the variance inflation factors. There were no missing data on the STAS-OA1 or any of the predictors.

IBM SPSS version 27 was used for all the analyses, all the tests were two-tailed, and we used a significance level of 0.05. ORs greater than 2.74 were considered to indicate a medium effect; ORs greater than 4.72 were considered to indicate a large effect [65].

## 3. Results

### 3.1. Occurrence of Reading Skills in Children with Down Syndrome Aged 6–8

At T1, the year the children started school, none of the participants were characterized as a reader based on the standardized decoding measure. However, parental data showed that 81.4% of the children recognized at least some written words at this age. An overview of the number of words the children recognized based on parental reports is presented in Table 1.

The words recognized were mainly the children’s own name, “mum” and “dad” or their family members’ first names.

At T2, after one year of school, the proportion of readers was 11.6% on the STAS-OA-1 (α = 0.86), and by 3rd grade (T3), it had increased to 18.6%.

As shown in Table 2, each reader read 2–10 words at T2, corresponding to the use of a logographic and/or phonological decoding strategy. At T3, the range had increased to 2–38 words. However, participant R3 was an outlier; this child’s score of 38 was consistent with the normal range for 4th grade (*M* = 39, *SD* = 23), corresponding to the use of an orthographic decoding strategy.

### 3.2. Differences in Cognitive Profiles between Readers and Non-Readers with Down Syndrome Prior to Reading Onset

The means and standard deviations for nonverbal mental ability, language and reading-related measures at T1 for both groups are shown in Table 3.

As shown in Table 3, the readers performed better than the non-readers on all the measures, with all but RAN showing medium-to-large effect sizes even after controlling for nonverbal mental ability. Differences in vocabulary and letter knowledge were significant after controlling for nonverbal mental ability. Additionally, there were significant bivariate group differences in nonverbal mental ability, short-term memory and phonological awareness. However, the group differences in RAN were not significant.

## 4. Discussion

This longitudinal study aimed to investigate the occurrence of reading skills in a Norwegian age cohort of children with Down syndrome at 1st (T1), 2nd (T2) and 3rd (T3) grade and to provide insight into early abilities, as these abilities may differ between children who do and do not develop early reading skills during this period. In particular, we were interested in which abilities are strengths in early readers prior to their reading onset. The data showed that the occurrence of reading skills was low but increased over the years. Vocabulary and letter knowledge were stronger in readers than in non-readers prior to their reading onset.

### 4.1. Occurrence of Reading Skills in a Norwegian Age Cohort of Children with Down Syndrome

According to the parental measure, the majority of the children were able to recognize some words at T1. None of the children could decode words on the standardized reading measure at this time point. The fact that children could mainly recognize names or a very limited set of words and were not able to decode words on a standardized decoding test might reflect that these children utilized a logographic strategy rather than having reached a phonological decoding level at this point in time. However, the occurrence of decoding increased over the years. By 2nd grade, 11.6% of the children with Down syndrome achieved measurable levels of decoding skills on the standardized measure, and this proportion increased to 18.6% of the sample by 3rd grade. This result may appear to be a low occurrence of reading skills compared to that of age-spread samples of individuals with Down syndrome from previous research (e.g., [22,23,24]). However, the apparent discrepancy in results may be associated with the following three factors. (1) The first is the methodological aspect of the present study. The inclusion of an entire national age cohort allowed for a relatively large sample (the largest possible *n* for this age group at a national level within the time frame of the current study period) and meant that all the children were the same age. As a result, no children entered the study with decoding skills on the standardized decoding measure; they all started to receive instruction in reading at the start of the study and received this instruction for a similar length of time in the study period. Apart from the requirement of being six years of age, having Norwegian as the first language and having no comorbidity of ASD, there were no selections made in the recruitment procedure, for example, no requirement of verbal skills in the children, no specifications regarding the area of the country, and no consideration of whether the children accessed specific support services. (2) The second is related to the Norwegian educational system. For example, the strong role of play in kindergarten in Norway [66] made it likely that systematic instruction in reading would not have been introduced to our study participants before they started school at six years of age. Due to the developmental profile of children with Down syndrome it usually takes longer to learn new skills compared to typically developing children (cf. [53]). It is therefore likely that the occurrence will gradually increase with age and length of training. This reasoning is also supported by our data since the occurrence of reading skills increased over the years. Similarly, previous longitudinal research on children with Down syndrome supports this reasoning; Laws and Gunn [24] found a large increase in the occurrence of reading skills from age 11 (33% of their participants) to age 16 (53% of their participants), which corresponds to an average increase of 4% per year. Assuming that the occurrence of reading continues at the same pace for each year children receive instruction in reading, our results after two years of school education can be considered relatively consistent with those of studies such as Laws and Gunn [24]. (3) The third is the available educational resources adapted for Norwegian students with Down syndrome. Norwegian is an infrequently used language with relatively few available materials and seminars for parents and teachers working specifically with children with Down syndrome, while previous studies on occurrence have usually been conducted in English-speaking countries (e.g., [22,23,24]), where Sue Buckley and her team made available reading materials for this group of children from a very early age along with seminars for their parents and teachers (Down Syndrome Education International https://www.down-syndrome.org/ (accessed on 18 April 2021)). Since the occurrence of decoding also varies greatly across previous research, the results of the present study complement earlier findings, applying data from school starters in a non-English-speaking country and using different inclusion criteria, measures and methodological approaches.

### 4.2. What Distinguishes Those Who Develop Early Decoding Skills from Those Who Do Not?

Given that individuals with Down syndrome tend to experience significant learning difficulties and considering the young age of the cohort of this study, it is not surprising that many children with Down syndrome are somewhat delayed in achieving phonological decoding skills compared to what is expected of their typically developing peers. However, our results indicate that there were significant differences in nonverbal mental ability between readers and non-readers, with those who developed early reading skills showing better nonverbal mental functioning. These results are in line with what is suggested both for typically developing children (c.f. [49]) and in a previous study of children with Down syndrome [24]. However, when the children’s nonverbal mental ability was controlled for, there were other variables that accounted for the differences between readers and non-readers prior to their reading onset. These findings demonstrate that the early reading ability of children with Down syndrome was not solely the result of stronger nonverbal mental ability. However, better nonverbal mental ability may have given these students access to reading interventions.

Prior to primary school education and reading onset, readers and non-readers displayed significant differences in vocabulary breadth and letter sound knowledge. The wide confidence interval of the vocabulary measure may limit the credibility of the odds ratio. However, both of these variables have been found to be reliable predictors of decoding in typically developing children (vocabulary [29]; letter sound knowledge [26]).

The importance of vocabulary is also consistent with earlier research on Down syndrome by Boudreau [46], Hulme et al. [19] and Steele et al. [67], who indicate that vocabulary is a stronger predictor of reading among these children than among typically developing children. As discussed by Hulme et al. [19], receptive vocabulary and expressive vocabulary tap into knowledge regarding both the phonological and semantic forms of words, which may help a child to both decode and develop contextual expectations about words to read in a concrete way. In line with this previous research, our findings indicate that early lexical knowledge may assist children with Down syndrome in obtaining decoding skills.

Moreover, we found that children with Down syndrome who exhibited word reading skills had greater letter knowledge than non-readers, as has also been observed in children with [68] and without Down syndrome (e.g., [69,70]). To understand the alphabetical principle and to use an analytic-based decoding approach, letter knowledge is necessary. As Muter et al. [69] have hypothesized, knowledge of the sounds of letters is also crucial for phonological decoding; that is, children understand that letter clusters represent phonemes. In addition, learning letter sounds provides a measure of paired visual–phonological associative learning that may correspond to the basic mechanism that is a fundamental component of learning to decode words [33]. The combination of good vocabulary and letter knowledge may help a reader to understand that words are made by letters, develop phonemic sensitivity [71] and predict words to read.

Finally, there is some indication that short-term memory and phonological awareness are underlying strengths in early readers with Down syndrome, as the effect sizes were substantial. However, due to lack of power when taking nonverbal mental abilities into account, the importance of these variables in children with Down syndrome learning to read are still inconclusive. Several previous studies have concluded that phonological awareness is reliably related to reading skills in children with Down syndrome (e.g., [36]); however, these studies measured the phonological measures post-reading onset. Based on the inconclusive results in the present study, we cannot interpret whether phonological skills are also important pre-reading onset. Thus, more studies are needed in future to clarify the role of phonological variables in the reading development of children with Down syndrome. It is also worth considering that reading development itself may promote phonological awareness, leading to differences in this outcome based on whether it is measured pre- versus post-reading onset [44,72].

### 4.3. Limitations and Strengths of the Study Design and Methods

This research is the first international longitudinal study investigating a relatively large age cohort of children with Down syndrome just starting school. In longitudinal studies, attrition usually occurs [73]. Nevertheless, in this study, no attrition occurred. We observed highly significant differences between readers and non-readers, which revealed clear early strengths among the readers. Future research is needed to determine whether each of these specific strengths of early readers plays a causal role in promoting reading development in children with Down syndrome. This study represents an important first step in identifying appropriate variables to investigate in future predictive studies.

Compared to other studies on reading skills in children with Down syndrome, the present study is robust in terms of its sample size; however, in regard to the power of the statistical analysis, the number of participants is still limited. To reduce the number of variables and possible bias of multiple comparisons or multicollinearity, we combined the scores of related predictors.

Children with Down syndrome may be slower at processing information than typically developing children, and therefore, the time frame for the standardized reading test may be challenging for them. However, we compared reading skills between two groups of children with Down syndrome, and all the subjects were likely to have the same processing problems. It could be argued that this reading test (the STAS test) underestimates the occurrence of reading skills in children with Down syndrome. However, this test is frequently used in Norwegian schools and is representative of how children’s decoding skills are usually measured among their typically developing peers. Additionally, similar standardized subtests were used in previous studies on occurrence (e.g., the Kaufman Assessment Battery for Children [24] and the Woodcock Reading Mastery Test [22]). Based on the slow reading progress and the fact that a relatively high percentage of children scored 0 correct answers on the standardized measure but scored higher on the parental reports, this situation may call for more sensitive and reliable measures designed specifically for children with Down syndrome in the future. These future measures should take the reading development process into account to detect small changes in children’s reading performance.

We did not collect specific data about reading interventions among the children, but the national curriculum for mainstream schools focuses heavily on phonological awareness and reading instruction for grades 1 and 2 (LK06). Where possible, all children, including those with Down syndrome, follow this curriculum.

## 5. Conclusions

This study demonstrates that children with Down syndrome can develop reading skills by 3rd grade. Our study extends the current literature by examining the age-specific occurrence of reading across the participants’ first three years of school in Norway and by highlighting variables that may be underlying strengths of early readers with Down syndrome. Specifically, early readers exhibited significant strengths in vocabulary and letter knowledge skills prior to their reading onset, in addition to stronger nonverbal mental ability. As the children were pre-readers when these data were collected, the strengths cannot be attributed to learning via reading. Thus, these findings reinforce the need to consider children’s language skills as well as their understanding of the alphabetical principle; early systematic training in vocabulary and play with letter knowledge could play a key role in promoting the development of early reading skills in children with Down syndrome.

## Figures and Tables

**Table 1 brainsci-11-00527-t001:** Overview of the number of words recognized by the children at T1 based on parental reports.

Number of Words the Child Can Read	*N*	Percentage
0 words	8	18.6%
1–5 words	20	46.5%
6–10 words	7	16.3%
11–15 words	2	4.7%
More than 15	3	7%
Unknown *	3	7%
Total	43	100%

* A parent reported that their child could recognize words, but the response to frequency was omitted.

**Table 2 brainsci-11-00527-t002:** Number of words on the STAS-OA1 read by each of the individual participants who were classified as readers at T1, T2 and T3.

Readers	T1	T2	T3
R1	0	2	9
R2	0	0	4
R3	0	10	38
R4	0	0	2
R5	0	0	11
R6	0	5	13
R7	0	4	6
R8	0	2	3

Note: The values represent the *number* of words read at each time point; R = reader.

**Table 3 brainsci-11-00527-t003:** Mean standardized (*Z*) scores (*SD*) and *OR* for the cognitive measures at T1 of readers vs. non-readers at T3.

	Readers (*n* = 8)	Non-Readers (*n* = 35)	Bivariate Analyses		Controlled for Nonverbal Mental Ability	
	*M* (*SD*)	*M* (*SD*)	*OR* (95% *CI*)	*p*-Value	*OR* (95% *CI*)	*p*-Value
Nonverbal mental ability	0.65 (1.14)	-0.15 (0.92)	2.85 (1.00–8.19)	0.05		
Vocabulary	0.95 (0.50)	−0.22 (0.85)	9.86 (1.88–51.75)	0.007	8.04 (1.46–44.26)	0.02
Verbal short-term memory	0.54 (0.79)	−0.12 (0.61)	3.94 (1.20–12.96)	0.02	3.01 (0.82–11.05)	0.10
Letter knowledge	1.01 (1.16)	−0.23 (0.81)	3.07 (1.38–6.84)	0.006	2.77 (1.22–6.29)	0.02
RAN	−0.29 (0.23)	0.07 (0.91)	0.53 (0.17–1.64)	0.27	0.33 (0.08–1.42)	0.14
Phonological awareness	0.56 (0.50)	−0.13 (0.85)	4.75 (0.97–23.26)	0.05	5.83 (0.77–44.09)	0.09

Note: The odds ratio (*OR*) with the 95% confidence interval (*CI*) is from logistic regression analyses. Predictors were standardized (*Z*) before being entered into the models. Higher scores reflect better performance on all the measures, except for the RAN task, in which lower scores reflect faster naming speed. Bold = significant group differences (*p* ≤ 0.05).

## Data Availability

The data are available in the services for sensitive data at the University of Oslo and can be obtained by contacting the first author.

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
