# Peer review of "Occurrence of Reading Skills in a National Age Cohort of Norwegian Children with Down Syndrome: What Characterizes Those Who Develop Early Reading Skills?"

_brainsci, 2021, doi:10.3390/brainsci11050527_

Round 1

Reviewer 1 Report

In this longitudinal study, researchers examined the earliest predictors of attaining decoding skills in a sample of children with Down syndrome (n=43) from first to third grade. Results indicated that nearly 19% of the participants demonstrated decoding skills by the beginning of third grade. Notably, children who developed decoding by grade three also demonstrated greater competence in vocabulary and letter knowledge upon entry to first grade when controlling for nonverbal mental age. They also demonstrated better short-term memory and phonological awareness bivariately.

Thank you for the opportunity to review this manuscript. Overall, the authors’ approach of examining variabilities in outcomes and the predictors of this variability is a strength of this manuscript and the results will be interesting to many readers. However, the manuscript would benefit from clarifications regarding methodology, results, and limitations. Notably, reporting effect sizes as opposed to predictive regressions is an inventive approach for this modest-sized sample. I request more information to better evaluate this aspect of the findings better. The below comments, organized by section, are intended to facilitate the publication of this manuscript.

INTRODUCTION

The examination of the earliest predictors of reading skills is a strength of this manuscript. Also, the first paragraph of this section Is an excellent introduction to the topic.

Section 1.3 The Occurrence of Reading- I expected the authors to succinctly comment on the variability of occurrences in reading among the cited studies  (e.g., small samples? Cohort effects?)

Line 136/137 it is “intellectual disability”

PARTICIPANTS- report how many families were invited in addition to the number who accepted the invitation

Typically, researchers also report more participant characteristics so that readers can interpret the generalizability such as co-occurring conditions (e.g., did any participants have hearing problems?), gender, race/ethnicity, SES mother’s highest level of education, years when data collection occurred, etc.

MEASURES

Succinctly include psychometric properties (reliability, validity, standardization) of the STAS OA-1, Nonverbal Mental Age, Vocabulary Tests, and Verbal Short-Term Memory

Also were z scores generated for Nonverbal Mental Age?

RESULTS/DISCUSSION

Reporting effect sizes based on the t-values of the linear regressions, as opposed to predictive regressions, is an inventive approach for this modest-sized sample. Was this an overall regression? Or individual regressions/correlations with each predictor and the outcome variable? Finally, help me understand why you chose this approach rather than correlations.

Related to the above comment, I am not certain it is accurate to state which variables had the strongest predictive value (e.g., line 451) given that it’s not clear that you did an overall regression. So, the use of language needs to match the approach.

Table 3,  please clarify in the table title/column headings that Readers/Non-Readers data is from T3, the left-hand column is difficult to read as formatted, there is no bold formatting for significant findings

Section 4.3 Limitations- the study has more limitations that need to be acknowledged (comparison group, participant characteristics largely unreported/unknown, sample size) and be sure to include the longitudinal design as a strength

MINOR EDITS

Lines 12-16, this is a long and sometimes hard to follow sentence, revise for readability

Line 18, since you introduce the time frame as first through third grade at the beginning of the abstract, then on line 18 it would be more helpful for the reader to refer to third grader or than 8 years of age

Line 106, please revise sentence starting on this line for readability

Line 111-117, please revise sentence for readability

Line 302/303, it is “two-tailed”

Line 374-378, please revise sentence for readability

Line 383-385, please revise sentence for readability

393-395, please revise sentence for readability

Reviewer 2 Report

The manuscript titled "Occurrence of Readers in a National Age Cohort of Norwegian Children with Down Syndrome: What Characterizes Those Who Develop Early Reading Skills?" described a longitudinal study investigating reading ability and contributing factors in Norwegian children with Down syndrome. A longitudinal study in this field is critically important, and this study is interesting. The participants were 43 children with Down syndrome. They were evaluated at 6 years of age and followed by 8yo.

I have several comments for the manuscript in its current form. The followings are my comments, observations, and suggestions:

-In this reviewer's understanding, this study aimed to detect factors affecting acquisition of the reading ability in children with Down syndrome. The outcome variable was the reading ability (occurrence of the readers: reader vs. non-reader) at T3 evaluated by the STAS-OA test. Binary logistic regression analysis predicting readers' occurrence is a more appropriate way to examine the cognitive variables affecting reading status at T3, instead of linear regression analysis.  

-As the authors described, intellectual disability is also an essential factor affecting reading ability in Down syndrome. Were the participants evaluated general intellectual function (e.g., mild to severe intellectual disabilities)? Most individuals with Down syndrome have moderate to severe intellectual disabilities, although it varies.

Minor comments

-Line 31, please support the statement with a reference.  

-Line 302, "to" should be "two."

-Line 309-312, please move to the methods section.

-It is not clear what Table 2 shows. Do the rows show individual participants who were the "readers" at T1-T3? Please clarify.

Round 2

Reviewer 2 Report

This reviewer appreciated the authors’ efforts to revise the manuscript. Now, I have a few comments for the added contents in the revised manuscript. The revision improved the clarity of the manuscript.

-Please be consistent with the reporting in methods, results, and discussion. For example, the logistic regression analyses showed the primary results about the associations between cognitive variables and reading onset after controlling for non-verbal mental ability. In contrast, the associations without adjustment were reported in the summary of the results in the first paragraph of the Discussion section. Such inconsistencies are required to be revised (particularly in the Discussion section).

-Although Vocabulary was significantly associated with the reading onset at T3, the confidence interval is so wide in the current sample, limiting the credibility of the odds ratio. This should be succinctly stated.

-As the authors described, the number of observations is limited in this study. Please avoid detailed interpretation based on non-significant results about verbal short-term memory and phonological awareness (page 11). Limited statistical power does not allow us to interpret the non-significance of the statistical analysis.

Minor comments

-In the abstract, remove line 20-21, the sentence about short-term memory and phonological awareness associations. Bivariate analysis is not the primary result of this study. 
